# *PTPN22* R620W gene editing in T cells enhances low-avidity TCR responses

Warren Anderson[1], Fariba Barahmand-pour-Whitman[2], Peter S Linsley[2], Karen Cerosaletti[2], Jane H Buckner[2], David J Rawlings[3]*

[1]Center for Immunity and Immunotherapies, Seattle Children's Research Institute, Seattle, United States; [2]Benaroya Research Institute at Virginia Mason, Seattle, United States; [3]Department of Pediatrics and Immunology, University of Washington, Seattle, United States

*For correspondence:
drawling@u.washington.edu

Competing interest: The authors declare that no competing interests exist.

**Abstract** A genetic variant in the gene *PTPN22* (R620W, rs2476601) is strongly associated with increased risk for multiple autoimmune diseases and linked to altered TCR regulation and T cell activation. Here, we utilize Crispr/Cas9 gene editing with donor DNA repair templates in human cord blood-derived, naive T cells to generate *PTPN22* risk edited (620W), non-risk edited (620R), or knockout T cells from the same donor. *PTPN22* risk edited cells exhibited increased activation marker expression following non-specific TCR engagement, findings that mimicked *PTPN22* KO cells. Next, using lentiviral delivery of T1D patient-derived TCRs against the pancreatic autoantigen, islet-specific glucose-6 phosphatase catalytic subunit-related protein (IGRP), we demonstrate that loss of PTPN22 function led to enhanced signaling in T cells expressing a lower avidity self-reactive TCR, but not a high-avidity TCR. In this setting, loss of PTPN22 mediated enhanced proliferation and Th1 skewing. Importantly, expression of the risk variant in association with a lower avidity TCR also increased proliferation relative to *PTPN22* non-risk T cells. Together, these findings suggest that, in primary human T cells, *PTPN22* rs2476601 contributes to autoimmunity risk by permitting increased TCR signaling and activation in mildly self-reactive T cells, thereby potentially expanding the self-reactive T cell pool and skewing this population toward an inflammatory phenotype.

## Editor's evaluation

Although the experimental system is limiting in addressing intracellular signaling parameters, the overall conclusions of the paper are important in that they further our understanding of the mechanisms through which the PTPN22 R620W variant, associated with multiple autoimmune diseases, contributes to breech of peripheral T cell tolerance. Moreover, this work greatly advances and clarifies ongoing confusion about whether PTPN22 SNP(620W) is a Loss Of Function mutant.

## Introduction

Preventing inappropriate immune responses is critical to the maintenance of peripheral tolerance and the prevention of autoimmunity. The risk of autoimmunity stems from a complex interplay of genetic and environmental factors (*Theofilopoulos et al., 2017*; *Cho and Feldman, 2015*). While the greatest risk factor for most autoimmune disease is HLA haplotype (*Todd et al., 2007*; *Bonifacio et al., 2018*), an individual's total genetic risk is determined by a combination of gene variants (*Sharma et al., 2018*; *Cotsapas et al., 2011*), many of which cluster near genes responsible for regulating immune responses (*Farh et al., 2015*). A leading example of such an autoimmune risk variant is found in the gene protein tyrosine phosphatase non-receptor 22 (*PTPN22*).

While the phosphatase encoded by *PTPN22* can function to modulate a range of immune signaling programs, it has been best studied in lymphocyte antigen receptor regulation (*Cohen et al., 1999*; *Wu et al., 2006*; *Hill et al., 2002*). An arginine to tryptophan substitution (R620W) resulting from a single nucleotide polymorphism (SNP; rs2476601) within *PTPN22* is broadly associated with a large number of autoimmune diseases (*Zheng et al., 2012*; *Tizaoui et al., 2019*). *PTPN22* R620W, hereafter referred to as the *PTPN22* risk variant, is one of the strongest, non-HLA, genetic risk factors for auto-immune diseases, nearly doubling a carrier's risk of type 1 diabetes, systemic lupus erythematosus, or rheumatoid arthritis (*Bottini et al., 2004*; *Begovich et al., 2004*; *Kyogoku et al., 2004*). As *PTPN22* plays a role in multiple autoimmune diseases, it has been argued that this SNP potentially highlights a common pathway that alters immune tolerance and might serve as a potential point of intervention to prevent and/or delay disease (*Rawlings et al., 2015*).

In T cells, PTPN22 functions to dephosphorylate key activating tyrosine residues within the prox-imal tyrosine kinase effectors, LCK and ZAP70, in the TCR signaling pathway (*Wu et al., 2006*; *Bottini et al., 2004*). This function is mediated in concert with activity through PTPN22's binding partner, C-terminal Src kinase (CSK), which also regulates LCK activity (*Cloutier and Veillette, 1999*). Co-asso-ciation of PTPN22 and CSK results in residence of both proteins within the cytosolic plasma membrane of resting T cells through the association of CSK with another TCR inhibitory protein, PAG (*Davidson et al., 2016*; *Simoncelli et al., 2020*). The R620W substitution in *PTPN22* is located within a proline rich, C-terminal domain and reduces the interaction of PTPN22 with CSK leading to altered localiza-tion of PTPN22 (*Bottini et al., 2004*). The functional outcome of disrupted PTPN22 protein interac-tions has led to significant debate as to the mechanism by which the *PTPN22* risk variant contributes to altered T cell function and self-tolerance, namely whether the human *PTPN22* risk variant mediates a gain- vs. a loss-of-function impact on TCR regulation.

Primary T cells isolated from carriers of the *PTPN22* risk variant display significantly weaker responses to TCR engagement, as measured by calcium flux, cytokine release, and other metrics (*Rieck et al., 2007*; *Vang et al., 2005*; *Arechiga et al., 2009*; *Vang et al., 2013*). However, in murine models, expression of the risk variant leads to enhanced TCR responses, as measured by the same metrics in addition to increased T cell proliferation and survival (*Zhang et al., 2011*; *Dai et al., 2013*; *Lin et al., 2016*). In contrast, genetic ablation of *PTPN22* in both mouse and human primary T cells results in enhanced TCR signaling relative to *PTPN22* wild type cells; findings similar to murine risk model yet opposite of the phenotypes described in human carriers of the risk variant (*Hasegawa et al., 2004*; *Anderson et al., 2019*). Furthermore, genetic ablation of *ptpn22* in a fixed TCR murine model has shown that loss of PTPN22 preferentially impacts T cells responses to low-avidity antigens, but not high-avidity antigens (*Salmond et al., 2014*); driving increased proliferation and skewing toward a Th1 phenotype in *ptpn22* KO T cells. While these observations suggest an important poten-tial mechanism for how the *PTPN22* risk variant promotes autoimmunity, this concept has not been tested in primary human T cells.

Both human carriers and murine models of the *PTPN22* risk variant share important traits. For instance, the risk variant is associated with an increase in memory lymphocyte populations and expanded pro-inflammatory/Th1 T cells and both exhibit increased risk of autoimmune manifestations (*Vang et al., 2013*; *Dai et al., 2013*; *Sanchez-Blanco et al., 2018*). Thus, it remains surprising that murine and human carriers exhibit these key similarities despite the published differences in the TCR signaling responses described human vs. mouse T cells. Notably, the murine and human T cell popu-lations used for previous studies exhibit fundamental differences. First, most data for human *PTPN22* risk variant phenotype have utilized mature T cells isolated from adult donors, while T cells from murine studies utilize donor mice with minimal environmental exposures. Studies of *PTPN22* using human cord blood-derived T cells have been limited (*Rawlings et al., 2015*). Also, murine models of the *ptpn22* risk variant are conducted in genetically homogenous, autoimmune-prone strains or models with a fixed TCR repertoire (*Dai et al., 2013*; *Lin et al., 2016*; *Salmond et al., 2014*; *Maine et al., 2016*), conditions difficult to mimic in a primary human T cell setting. Further, in vitro human T cell line models of the *PTPN22* risk variant lead to discrepant results, indicating that the impact(s) of the variant are context dependent, further complicating efforts to study the variant in human T cells (*Vang et al., 2005*; *Zikherman et al., 2009*).

Recently, Crispr/Cas9 gene editing has allowed for precise manipulation of genetic loci in primary human T cells. Delivery of recombinant ribonucleoproteins (RNPs [*Anderson et al., 2019*; *Schumann*

*et al., 2015*; *Okamoto et al., 2019*] containing Cas9 complexed with sequence specific guide RNA sequences) via electroporation into primary T cells efficiently mediates site-specific DNA double stranded breaks (DSBs) near a protospacer adjacent motif (PAM) site (*Jinek et al., 2012*). Such DNA breaks trigger a series of events driving 5' to 3' exonuclease activity at the broken DNA ends as the cell prepares to mend the break through potential repair pathways (*Scully et al., 2019*; *Xue and Greene, 2021*). When DSBs are generated in the presence of a single stranded oligo deoxynucleotides (ssODNs) (co-delivered with RNP), DNA repair pathways can utilize regions of homology within an ssODN as a repair template for precise genetic modification by homology directed repair (HDR) (*Richardson et al., 2016*; *Richardson et al., 2018*). This approach has previously been used to integrate a premature stop codon in the *PTPN22* locus of primary human T cells with high efficiency (*Anderson et al., 2019*).

In this study, we utilized a combination of HDR-based gene editing and TCR co-delivery in naïve cord blood T cells to establish a platform to better understand the role of the *PTPN22* risk variant in primary human T cells. Through Crispr/Cas9 gene editing we find the *PTPN22* risk variant T cells mimicked *PTPN22* KO T cells, with enhanced expression of surface activation markers in response to TCR stimulation. Further, using transgenic TCR stimulation in concert with gene edited cells, we found that the *PTPN22* risk variant accentuates the response of low-avidity primary human T cells to antigen, implying a key role for these events in the progressive loss of T cell tolerance in human autoimmune diseases associated with this risk allele.

## Results
### Generation of PTPN22 risk and non-risk edited T cells with Crispr/Cas9 gene editing

In this study, we sought to identify methods to utilize HDR-based gene editing to study alternative *PTPN22* alleles within identical populations of primary human T cells. Notably, previous work has shown that altering expression levels of *PTPN22* and/or its regulatory proteins can produce potentially contradictory results (*Vang et al., 2005*; *Zikherman et al., 2009*). Additionally, *PTPN22* contains several splice variants that might modulate the impact of the risk variant (*Ronninger et al., 2012*). Therefore, we sought to utilize an editing platform that would permit us to introduce either a control, risk, or knockout (KO) sequence via direct modulation of the endogenous SNP locus, without the addition of exogenous promoters or selection agents that may alter natural *PTPN22* expression. To accomplish this goal, we expanded on previously reported methods to generate *PTPN22* gene disrupted primary human T cells (*Anderson et al., 2019*) to establish an HDR-based editing platform. We combined delivery of Crispr/Cas9 with ssODN HDR templates designed to alter the coding region of exon 14 of *PTPN22*. Use of Crispr/Cas9 RNPs with short ssODN repair templates to alter the DNA of primary cells has been shown to be highly efficient, provided the nuclease cuts near the desired edit site (*Meitlis et al., 2020*). A naturally existing Crispr/Cas9 gRNA PAM site is positioned in exon 14 of *PTPN22* to allow an RNP induced DSB immediately 5' of the SNP nucleotide (*Figure 1A*). Also, previous work has shown that homology arm design of ssODN repair templates may impact editing rates when used with a Crispr/Cas9 RNP (*Richardson et al., 2016*). We tested several ssODN repair template designs with our *PTPN22* RNP for optimized introduction of a 2 bp coding alteration to generate the *PTPN22* risk variant (*Figure 1A*; *Figure 1—figure supplement 1A*). Importantly, this 2 bp edit blocks subsequent binding of the RNP, preventing cleavage of the repaired locus. We found that in adult peripheral blood CD4 T cells, a 120 bp ssODN design with a long 5' homology arm (ssODN V3, 5' arm 89 bp/3' arm 29 bp) efficiently introduced the desired coding sequence alterations (*Figure 1—figure supplement 1B*), in line with previous findings (*Richardson et al., 2016*). Using this design, we created three ssODN repair templates that worked efficiently in association with the *PTPN22* RNP to produce three gene editing outcomes: non-risk edited *PTPN22*, risk edited *PTPN22*, and a *PTPN22* KO ssODN that seamlessly introduces a stop codon into all three reading frames of the RNP cut site (*Figure 1A*).

Natural carriers of the *PTPN22* risk variant have altered peripheral T cell responses, and mouse models indicate that variant expression impacts thymic T cell selection as well as peripheral T cell activation and function (*Dai et al., 2013*). Thus, human T cells isolated from subjects with the risk variant are likely to be previously impacted by diverse developmental, age-related and environmental

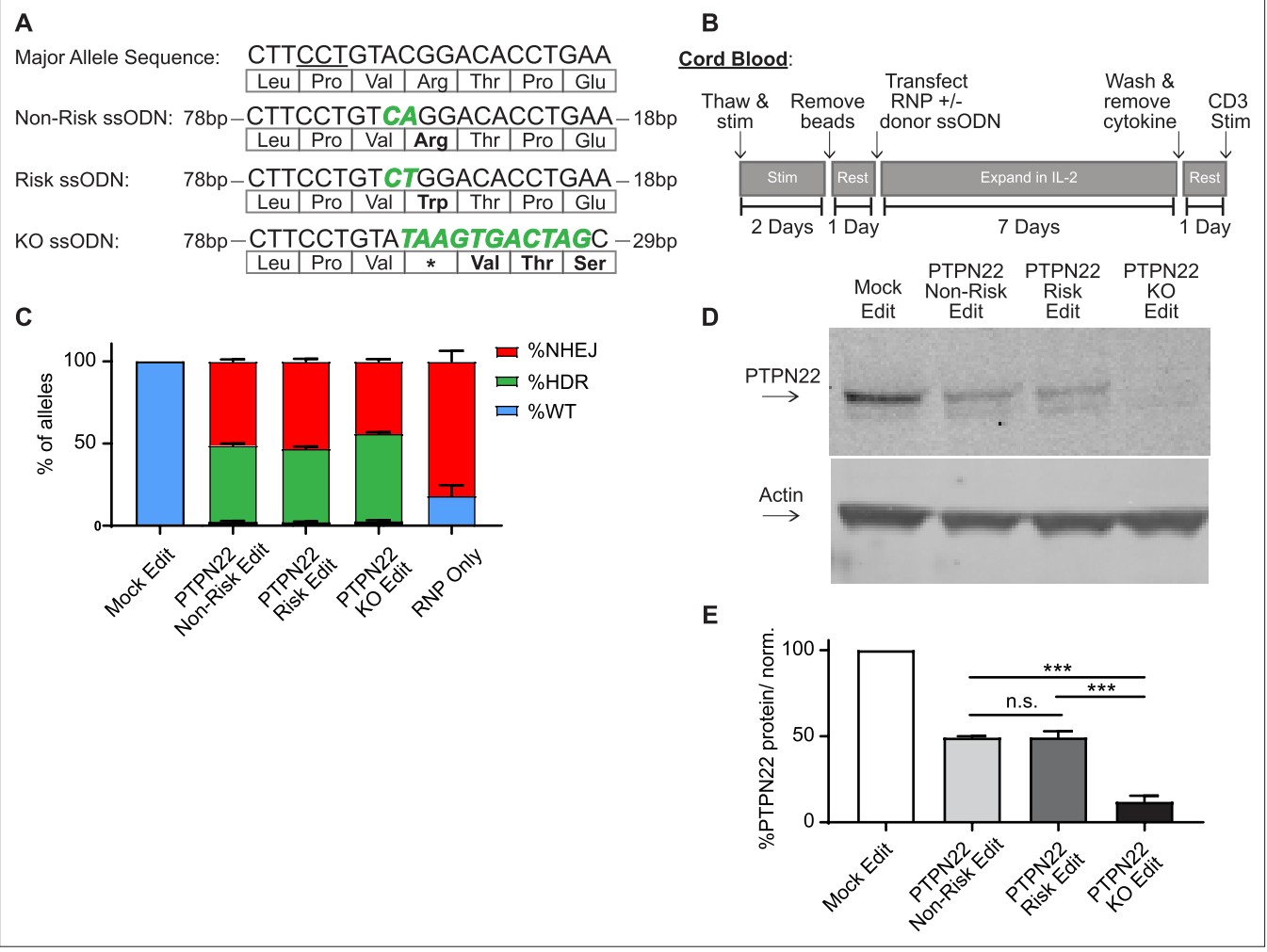

**Figure 1.** Gene editing protein tyrosine phosphatase non-receptor 22 (*PTPN22*) single nucleotide polymorphism (SNP) variants in cord blood CD4 T cells produces comparable populations. CD4 T cells were isolated from human umbilical cord blood for Crispr/Cas9 gene editing. (**A**) ssODN repair templates electroporated with *PTPN22* targeting Crispr/Cas9 RNP resulting in differently coded alleles upon HDR. Underlined nucleotides in major allele sequence indicate RNP protospacer adjacent motif (PAM) site. *PTPN22* coding alterations for each population denoted with bolded/colored nucleotides, resulting changes to amino acid sequence displayed below. (**B**) Workflow used to produce and assay *PTPN22* edited CD4 T cells and corresponding controls. (**C**) ddPCR analysis of editing outcomes in *PTPN22* edited cells using corresponding ssODN repair templates or RNP alone. n=4 independent human donors. (**D**) Representative western blot of PTPN22 expression in mock edited, *PTPN22* non-risk edited, *PTPN22* risk edited, and *PTPN22* knockout edited cord blood CD4 T cells from the same human donor. (**E**) Quantified PTPN22 protein expression relative to actin and normalized to unedited values from the same T cell donor. n=3 independent human donors, matched one-way ANOVA with Tukey's correction. For summary graphs, lines and error bars represent mean ± SEM. RNP – ribonucleoprotein, ODN or ssODN – single stranded oligo deoxynucleotide, NHEJ – non-homologous end joining, HDR – homology directed repair. All data is from at least two independent experiments. ***p<0.001.

The online version of this article includes the following source data and figure supplement(s) for figure 1:

**Source data 1.** Whole western blots.

**Source data 2.** Protein tyrosine phosphatase non-receptor 22 (PTPN22) gene editing assessment.

**Figure supplement 1.** Gene editing efficiency of protein tyrosine phosphatase non-receptor 22 (*PTPN22*) single nucleotide polymorphism (SNP) variants is impacted by donor DNA template design.

**Figure supplement 1—source data 1.** Protein tyrosine phosphatase non-receptor 22 (PTPN22) gene editing optimization.

exposures (*Rawlings et al., 2015*). Therefore, we reasoned that using naïve, cord blood-derived CD4 T cells isolated from non-risk human donors, which are minimally impacted by previous environmental events, would provide the most relevant cell population for our study and generate the most informative dataset with respect to impact(s) of the variant on human T cell function. Using our *PTPN22* RNP and ssODN repair templates with a gene editing workflow (*Figure 1B*), we edited

human cord blood-derived naïve CD4 T cells from multiple *PTPN22* non-risk donors to concurrently model the *PTPN22* risk variant, control edited, and KO edited cells. Comparing edited populations to mock edited cells (which received Cas9 without gRNA and *PTPN22* KO ssODN), all gene edited populations showed similar viability, regardless of editing reagents used, with a slight reduction in viability observed in all T cell populations receiving electroporation (*Figure 1—figure supplement 1C*). Droplet digital PCR (ddPCR) analysis demonstrated nearly identical gene editing rates in cells edited with the *PTPN22* non-risk vs. risk ssODN (*Figure 1C*). Both edited populations exhibited average HDR rates of ~50%, and no detectable editing was present in mock edited populations. HDR rates in *PTPN22* KO edited cells were slightly higher, likely due to a lack of nucleotide mismatches between the ssODN template and the digested *PTPN22* locus. As expected, beyond HDR events, NHEJ events accounted for most of the remaining editing outcomes in cells modified in the presence of ssODNs. NHEJ events were absent or dominant, in mock and RNP-only edited populations, respectively. Western blot analysis confirmed that, *PTPN22* non-risk and risk edited populations retained equivalent expression levels of *PTPN22* (with a reduction in total PTPN22 protein compared with mock edited cells reflecting the NHEJ-mediated loss of protein expression in the absence of successful HDR). As expected, *PTPN22* KO populations exhibited near total absence of protein expression (*Figure 1D–E*).

Wild type *PTPN22* alleles were nearly undetectable in edited populations (<1% of total alleles), indicating that resulting genotypes of edited T cells can be assumed to a mixture of biallelic HDR, biallelic NHEJ (*PTPN22* KO), or HDR and NHEJ on opposite alleles (hemizygous). RNP based editing can induce large deletions that may bias sequencing of editing outcomes (*Kosicki et al., 2018*), making accurate single cell profiling challenging. As HDR rates in our approach were identified using probe based ddPCR detection, this data is resistant to this type of bias. With ~50% HDR rates and ~50% NHEJ rates, the edited populations would, at a minimum, be 50% biallelically edited and 50% full *PTPN22* KO. Of note, previous single cell studies in primary cell types (*Park et al., 2019*; *Mocciaro et al., 2018*) including primary T cells (*Mocciaro et al., 2018*), demonstrate that monoallelic HDR is far more frequent than biallelic HDR. Therefore, it is reasonable to assume the predominant editing outcome in our studies is hemizygosity with an undetermined frequency of biallelic editing events. However, as an independent edited control population is utilized in all our analyses, the effects of *PTPN22* KO alleles (monoallelic or biallelic) are accounted for when *PTPN22* risk edited cells are compared to non-risk edited populations. This control assures that our findings regarding the *PTPN22* risk variant can be attributed to the expression of differently coded *PTPN22* variants, and not differences caused by disruption of *PTPN22* alleles.

## PTPN22 risk edited T cells mimic PTPN22 KO cells

Following generation of T cells expressing equivalent levels of the *PTPN22* risk, non-risk, or *PTPN22* KO T cells from the same donor, we tested how alterations in *PTPN22* coding impacted responses to TCR stimulation. After gene editing and expansion, T cells were stimulated with plate bound anti-CD3 for 6–48 hr and sequentially evaluated for expression levels of key surface activation markers. As all *PTPN22* edits produced some loss of *PTPN22* expression, and PTPN22 is responsible for restraining TCR activation, it was unsurprising to find that mock edited cells had significantly less expression of the activation markers CD69, CD25, and CD71 compared with gene edited populations (*Figure 2A–C*). However, it was striking that, while *PTPN22* KO populations expressed the highest levels of these activation markers, the *PTPN22* risk edited cells, in comparison with non-risk edited cells, also exhibited increased expression of CD25 and CD71 at 48 hr post stimulation and a trend toward increased CD69 expression as early as 6 hr post stimulation (*Figure 2A–C*). Together, this data strongly suggests that TCR signaling strength is increased in cells expressing the *PTPN22* risk variant, and, that this phenotype is most similar to *PTPN22* KO T cells.

The *PTPN22* risk variant has been reported to modulate CD4 T cell populations, with some studies suggesting increased proportions of Th1 (*Vang et al., 2013*; *Salmond et al., 2014*; *Sanchez-Blanco et al., 2018*) populations in risk carriers. To examine if an altered *PTPN22* status is sufficient to modulate CD4 T cell skewing outcomes, *PTPN22* edited cord blood T cells were cultured with plate bound anti-CD3, soluble anti-CD28, and IL-12 to produce Th1 CD4 T cells. Three days after stimulation we found most cells producing Th1 associated cytokines, however no differences in the number of cells producing IFNγ or IL-2 were observed based upon edits made to *PTPN22* (*Figure 2—figure*

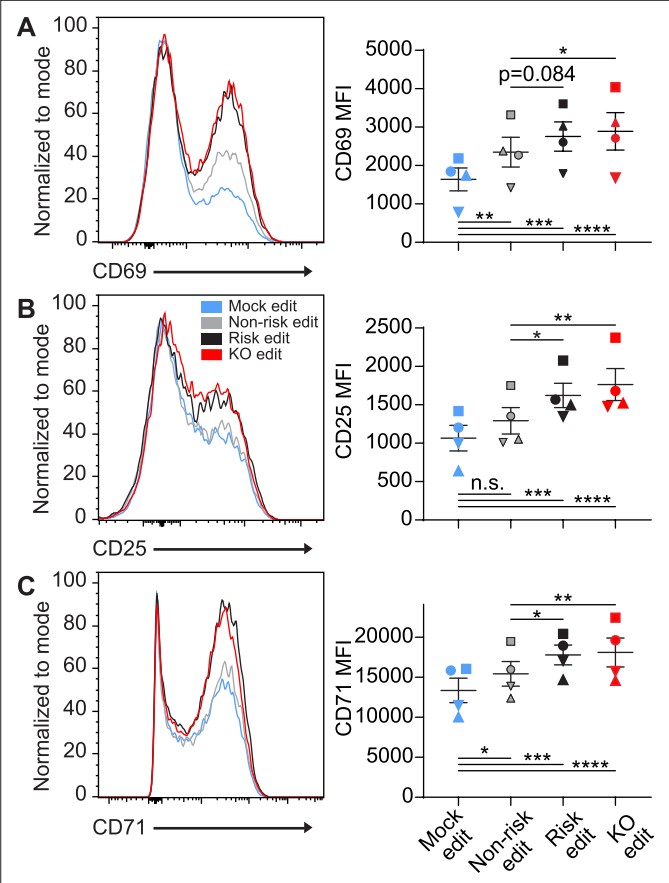

**Figure 2.** Protein tyrosine phosphatase non-receptor 22 (*PTPN22*) risk edited T cells mimic *PTPN22* knockout (KO) in response to TCR stimulation. Cord blood CD4 T cells from human donors were gene edited as described in *Figure 1*, then stimulated with plate bound anti-CD3 for up to 48 hr. (**A–C**) Representative flow overlays for CD69 (**A**), CD25 (**B**), and CD71 (**C**) with summary data for different editing conditions from the same donor. CD69 summary data reflects mean fluorescent intensity at 6 hr post CD3 stimulation, while CD25 and CD71 represent median values at 48 hr post stimulation. For all summary data n=4 independent donors (shapes correspond to individual donors), matched one-way ANOVA with Tukey's correction. Lines and error bars represent mean ± SEM. All data is from two independent experiments. *p<0.05, **p<0.01, ***p<0.001, ****p<0.0001.

The online version of this article includes the following source data and figure supplement(s) for figure 2:

**Source data 1.** CD3 stim, protein tyrosine phosphatase non-receptor 22 (PTPN22), edited cells.

**Figure supplement 1.** Protein tyrosine phosphatase non-receptor 22 (*PTPN22*) editing in T cells does not impact IL-12 induced Th1 skewing.

**Figure supplement 1—source data 1.** Protein tyrosine phosphatase non-receptor 22 (PTPN22) gene editing does not impact cytokine skewing.

*supplement 1A–B*). This lack of difference in Th1 skewing is consistent with findings in a similar study using the murine *ptpn22* risk variant model (*Sanchez-Blanco et al., 2018*).

## PTPN22 regulates signaling from low-avidity TCRs in human T cells

PTPN22 is a key regulator of TCR signaling and in murine models it plays a critical role in repressing T cell activation mediated by weak TCR agonists, but has been found to be dispensable in regulating responses to strong TCR agonists (*Salmond et al., 2014*). These findings suggested that perturbations in PTPN22 activity might impact autoimmune risk via enriching T cells expressing low-avidity self-reactive TCRs that escaped deletion during thymic development and remain capable of recognizing self-antigens. Thus, as our gene editing model indicated that the *PTPN22* risk variant produces a phenotype similar to a *PTPN22* KO, we next examined whether *PTPN22* KO modified the response of human T cells expressing TCRs with differing avidity for an identical self-antigen.

Previous work has detailed the sequencing of multiple TCRs from T1D patients that are reactive against the pancreatic autoantigen, islet-specific glucose-6 phosphatase catalytic subunit-related protein, IGRP (*Cerosaletti et al., 2017*; *Linsley et al., 2021*). Recent work has also shown that gene edited primary human T cells can be rendered antigen specific through the lentiviral delivery of TCRs prior to electroporation with RNPs (*Yang et al., 2021*). To study the impact of PTPN22 loss of function on antigen specific TCR responses, we transduced human cord blood CD4 T cells with lentiviral vectors (LV) encoding for alternative TCRs that respond to the same 20mer epitope of IGRP when presented by HLA-DRB1 04:01; and subsequently modified these LV transduced populations using HDR-based gene editing (*Figure 3A*). Addition of LV transduction prior to gene editing did not impact gene editing outcomes or rates of HDR (*Figure 3—figure supplement 1A*). Transgenic TCR chains were engineered to utilize the murine constant regions allowing staining with murine TCR specific antibodies (*Cerosaletti et al., 2017*) and limiting potential cross pairing with endogenous TCR chains. To track LV transduced cells, constructs were modified to express cis-linked GFP simultaneously with TCR expression. LV transduction was performed using a multiplicity of infection (MOI) of 5 in an effort to transduce <25% of the T cell population. This approach led to an average of 5–15% marking (depending on donor and LV product), ensuring low viral copy number per transduced cell and similar levels of TCR expression within experimental groups. After gene editing, expansion, and rest, gene edited/TCR+ T cells were stimulated via addition of irradiated PBMCs from an allogeneic HLA-DRB1 04:01 donor that had been pre-incubated with cognate IGRP peptide (*Figure 3B*). Following stimulation of TCR+ cord blood T cells with peptide loaded APCs, the alternative self-reactive TCRs responded to IGRP peptide at markedly different magnitudes. The two TCRs, that we term: 'lower avidity TCR' or L-TCR, and 'higher avidity TCR' or H-TCR, triggered different rates of proliferation over 3 days (based on either dye dilution or direct cell counts), consistent with their reported avidity (*Figure 3C and D*; *Linsley et al., 2021*). Notably, upon combining antigen specific stimulation with *PTPN22* KO gene editing, we found that L-TCR+ *PTPN22* KO cells had significantly increased proliferation relative to mock edited L-TCR cells across a range of peptide doses (*Figure 3C–E*). In contrast, H-TCR + expressing edited T cells showed similar proliferation rates in response to multiple antigen doses, regardless of *PTPN22* genotype (*Figure 3D and F*; *Figure 3—figure supplement 2A*). As the highest peptide dose (25 ng/ml) reliably produced efficient proliferation in all donors tested, this dose was used for further assessments.

To examine how loss of PTPN22 enhanced L-TCR induced proliferation, we assessed TCR-mediated, downstream phosphorylation events via flow cytometry. Unfortunately, stimulation of cells through antigen presentation in this model proved incompatible with examining early phosphorylation events (in contrast to stimulation using non-antigen-specific methods). T cells stimulated via peptide loaded APCs showed no consistent increase in pErk 1/2, pAKT, or total pTyr compared to cells stimulated with non-loaded APCs over the first 6 hr of co-culture (data not shown). However, intracellular signaling differences were present at 24 hr post peptide stimulation. In cells expressing L-TCR, the loss of PTPN22 resulted in significantly greater proportion of phosphorylated S6 kinase positive cells, while cells expressing H-TCR, *PTPN22* KO showed no impact on the relative percentage of pS6+ cells (*Figure 3G*).

We next determined if the early signaling differences found in *PTPN22* KO cells expressing L-TCR (vs. H-TCRs) led to differences in activation phenotype. Two days post peptide stimulation, L-TCR expressing *PTPN22* KO T cells, in comparison to mock edited T cells from the same cord blood donor also expressing L-TCR, expressed higher levels of activation markers including CD69, CD25, and CD71 (*Figure 4A–C*). In contrast, H-TCR expressing *PTPN22* KO T cells exhibited no differences in comparison to control T cells. Together, these results demonstrate that loss of PTPN22 activity preferentially permits enhanced signaling in response to low-avidity TCR engagement in primary human T cells.

To further explore the impact of *PTPN22* editing in this setting, we generated *PTPN22* mock, control edited, risk edited, and KO cells (as in *Figures 1 and 2*) expressing the L-TCR construct. After 24hr of peptide stimulation, we observed a significant increase in pS6 kinase and similar trend in p4E-BP1 in *PTPN22* KO cells (*Figure 3—figure supplement 2B–C*). However, while we observed a similar trend in *PTPN22* risk edited cells, this data did not reach statistical significance (*Figure 3—figure supplement 2B–C*); suggesting this approach was not adequate to quantify subtle signaling changes in response to cognate antigen.

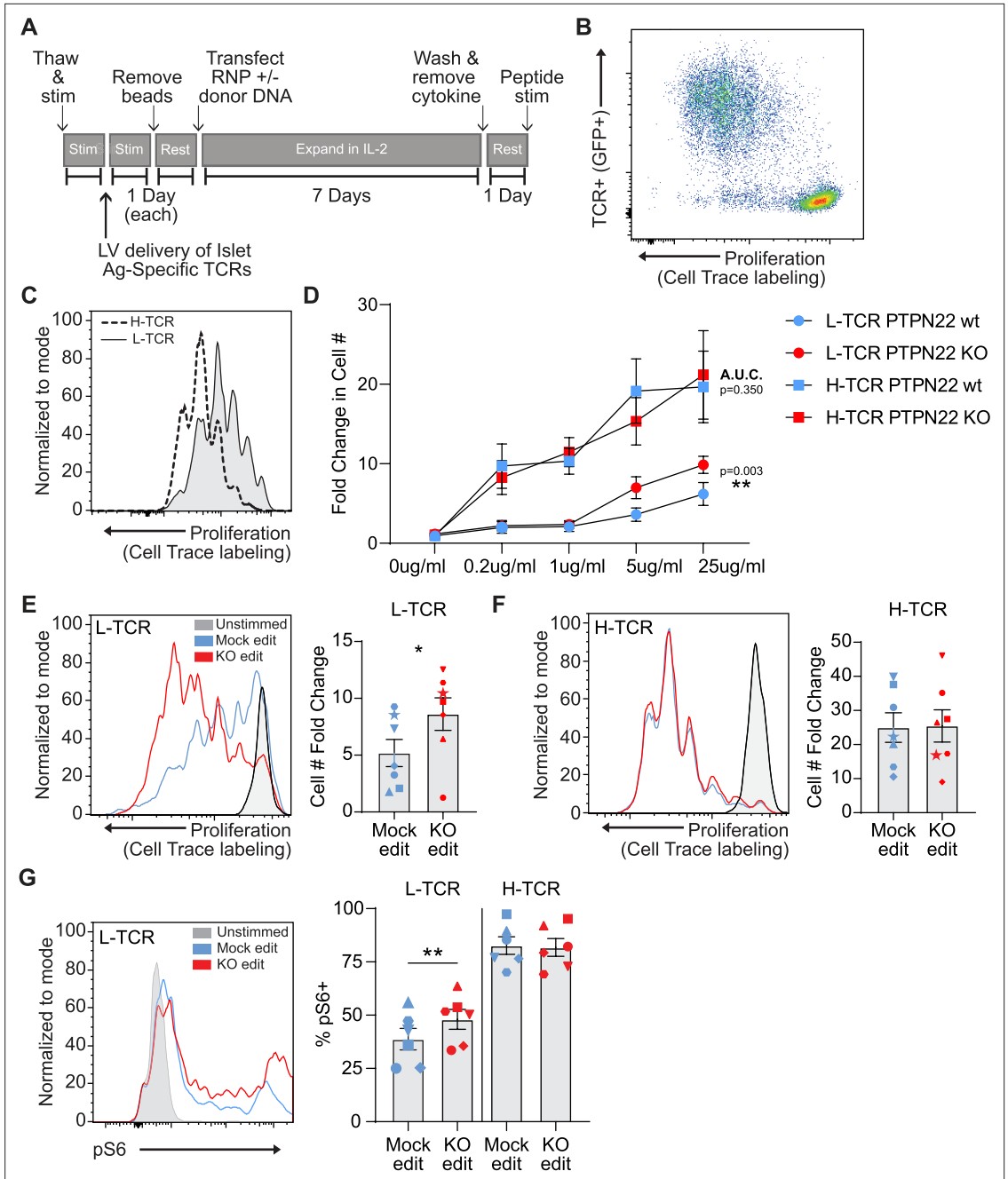

**Figure 3.** Low-avidity T cells lacking protein tyrosine phosphatase non-receptor 22 (PTPN22) exhibit increased proliferation upon peptide stimulation. Cord blood CD4 T cells from human donors were transduced with lentivirus coding for antigen specific TCRs prior to gene editing as described in *Figure 1*, then stimulated with cognate peptide. (**A**) Workflow shown in *Figure 1*, modified to allow both gene editing of the *PTPN22* locus and lentiviral delivery of antigen specific TCRs. (**B**) Representative flow plot of H-TCR+ T cell proliferation after 3 days of peptide stimulation. (**C**) Representative flow overlay of 3-day peptide induced proliferation caused by either H-TCR stimulation or L-TCR stimulation in T cells from the same donor. (**D**) Summary data of 3-day T cell proliferation (fold change over input cell number) at various peptide doses in cells expressing transgenic TCRs with and without *PTPN22* knockout (KO) editing. Proliferation induced by TCRs was compared by computing area under the curve (AUC) for each TCR/editing combination for all peptide doses tested, then performing a paired t test on the computed areas. (**E**) Representative flow overlay of proliferation in L-TCR+/ *PTPN22* gene edited cells from the same donor and summary data of T cell proliferation (fold change over input cell number), n=7 independent donors (shapes correspond to individual donors). (**F**) Representative flow overlay of proliferation in H-TCR+/ *PTPN22* gene edited cells from the same donor and summary data of T cell proliferation (fold change over input cell number), n=7 independent donors (shapes correspond to individual donors). (**G**) Representative flow overlay of different editing conditions from the same donor expressing L-TCR and summary data of flow values for phosphorylated S6 kinase in edited cells expressing transgenic TCRs, n=6 independent donors (shapes correspond to individual donors), paired t

*Figure 3 continued on next page*

*Figure 3 continued*

test by TCR expressed. Columns and error bars represent mean ± SEM. All data is from three or four independent experiments. *p<0.05, **p<0.01, ***p<0.001.

The online version of this article includes the following source data and figure supplement(s) for figure 3:

**Source data 1.** Peptide stim model of protein tyrosine phosphatase non-receptor 22 (PTPN22) edited cells.

**Figure supplement 1.** Protein tyrosine phosphatase non-receptor 22 (*PTPN22*) gene editing in cord blood CD4 T cells is not impacted by lentiviral transduction.

**Figure supplement 1—source data 1.** Gene editing does not impact lenti transduction.

**Figure supplement 2.** Protein tyrosine phosphatase non-receptor 22 (*PTPN22*) knockout (KO) cord blood CD4 T cells show increased proliferation and phosphorylation of activation responsive signaling markers.

**Figure supplement 2—source data 1.** Impact of altered protein tyrosine phosphatase non-receptor 22 (PTPN22) on early phosphorylation signals.

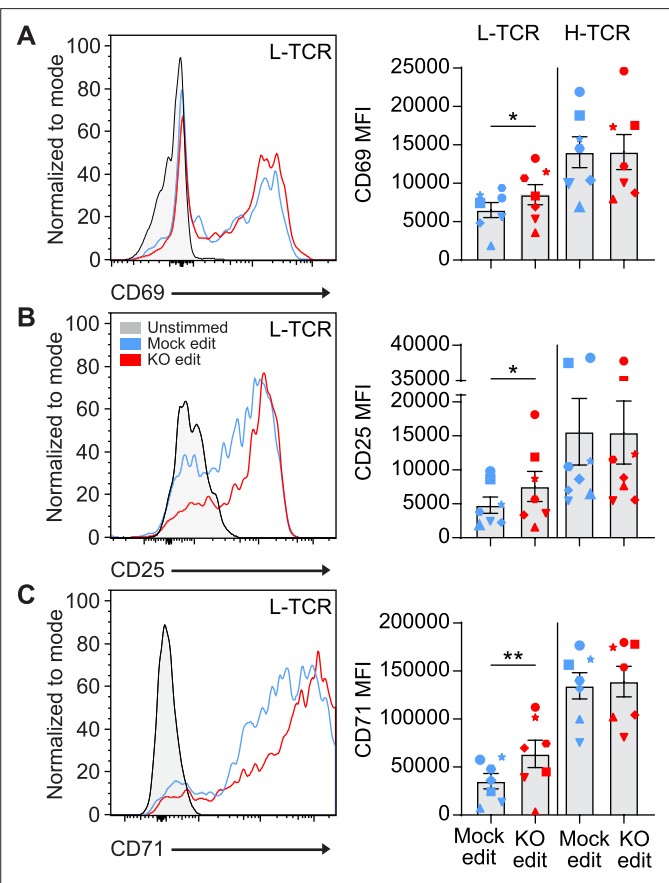

**Figure 4.** L-TCR stimulation of protein tyrosine phosphatase non-receptor 22 (*PTPN22*) knockout (KO) edited T cells produces enhanced surface activation marker expression. Cord blood CD4 T cells from human donors were edited as described in *Figure 3*, then stimulated with peptide loaded APCs for up to 48 hr. (**A–C**) Representative flow overlays and summary data for CD69 (**A**), CD25 (**B**), and CD71 (**C**) expression in mock edited and *PTPN22* KO populations from the same donor expressing the L-TCR. CD69 summary data reflects mean fluorescent intensity at 24 hr post peptide stimulation, while CD25 and CD71 represent median values at 48 hr post stimulation. For all summary data n=7 independent donors (shapes correspond to individual donors), paired t test by TCR expressed. Columns and error bars represent mean ± SEM. All data is from four independent experiments. *p<0.05, **p<0.01.

The online version of this article includes the following source data for figure 4:

**Source data 1.** Peptide stim activation response, protein tyrosine phosphatase non-receptor 22 (PTPN22) edited cells.

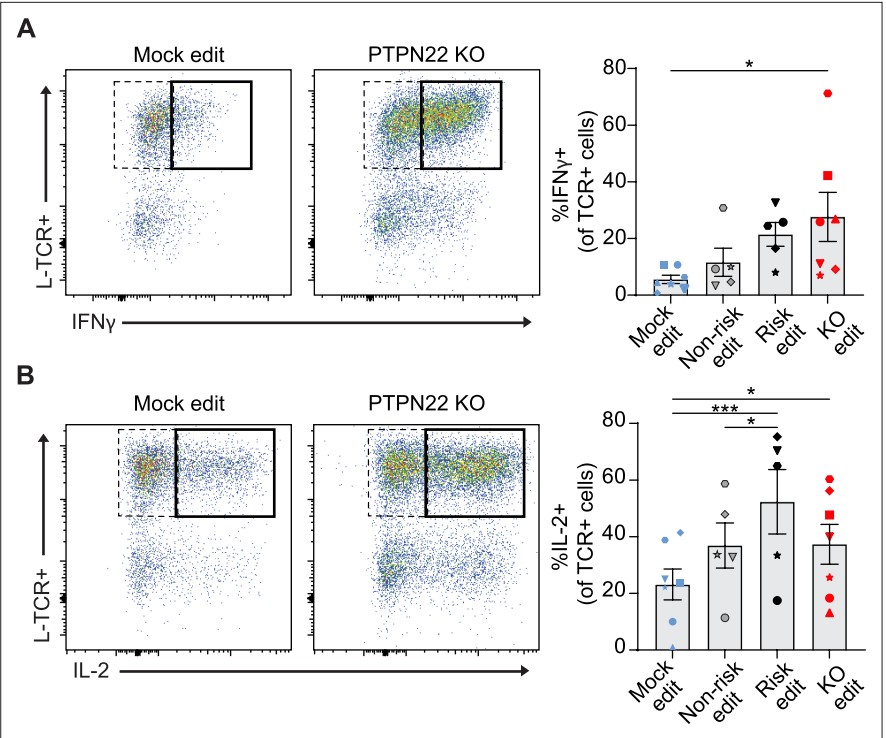

**Figure 5.** Protein tyrosine phosphatase non-receptor 22 (*PTPN22*) knockout (KO) and risk edited T cells are more likely to secrete pro-inflammatory cytokine in response to low-avidity TCR engagement. Cord blood CD4 T cells from human donors were edited as described in *Figure 3*, then stimulated with peptide loaded APCs for 3 days. (**A–B**) Representative flow plots of IFNγ (**A**) and IL-2 (**B**) expression in mock edited or *PTPN22* KO cells transduced with L-TCR and summary data of %L-TCR+ cells that secrete IFNγ (**A**) or IL-2 (**B**) grouped by gene edit. N=5–7 independent donors (shapes correspond to individual donors). All summary data analyzed by paired t test. Columns and error bars represent mean ± SEM. All data is from four independent experiments. *p<0.05, ***p<0.001.

The online version of this article includes the following source data and figure supplement(s) for figure 5:

**Source data 1.** Peptide stim cytokine production, protein tyrosine phosphatase non-receptor 22 (PTPN22) edited Cells.

**Figure supplement 1.** Protein tyrosine phosphatase non-receptor 22 (*PTPN22*) gene edited, H-TCR+ cord blood CD4 T cells show no differences in inflammatory cytokine production.

**Figure supplement 1—source data 1.** Protein tyrosine phosphatase non-receptor 22 (PTPN22) knockout (KO) does not impact cytokine production in strong TCR stim.

## PTPN22 deficiency promotes proliferation and IFNγ secretion by low-avidity T cells

A common feature of autoimmunity is inappropriate generation of pro-inflammatory cytokines, that drive diverse pathologies in various tissues. While PTPN22 loss had no impact on generation of an IL-12 driven Th1 phenotype in association with CD3/CD28 engagement (*Figure 2—figure supplement 1A–B*), we next tested whether *PTPN22* editing might impact the generation of a Th1 phenotype in association with TCR stimulation. We transduced and edited cord blood CD4 T cells, stimulated the edited populations with peptide loaded APCs for 3 days, and assessed the production of IFNγ and IL-2 by intracellular flow cytometry. We restricted our analysis to cells that had proliferated in response to stimulation to ensure the data was not impacted by changes in proliferation rates induced by *PTPN22* editing (*Figure 3D*). H-TCR+ *PTPN22* KO T cells exhibited no difference in frequency of IFNγ or IL-2 producing cells relative to mock edited cells (*Figure 5—figure supplement 1A-C*). In contrast, the proportion of L-TCR+ cells that generated IFNγ were increased by loss of PTPN22 and risk edited cells exhibited a similar trend though not significant (*Figure 5A*). Notably, *PTPN22* risk edited cells contained a significantly higher proportion of IL-2 producing cells (*Figure 5B*).

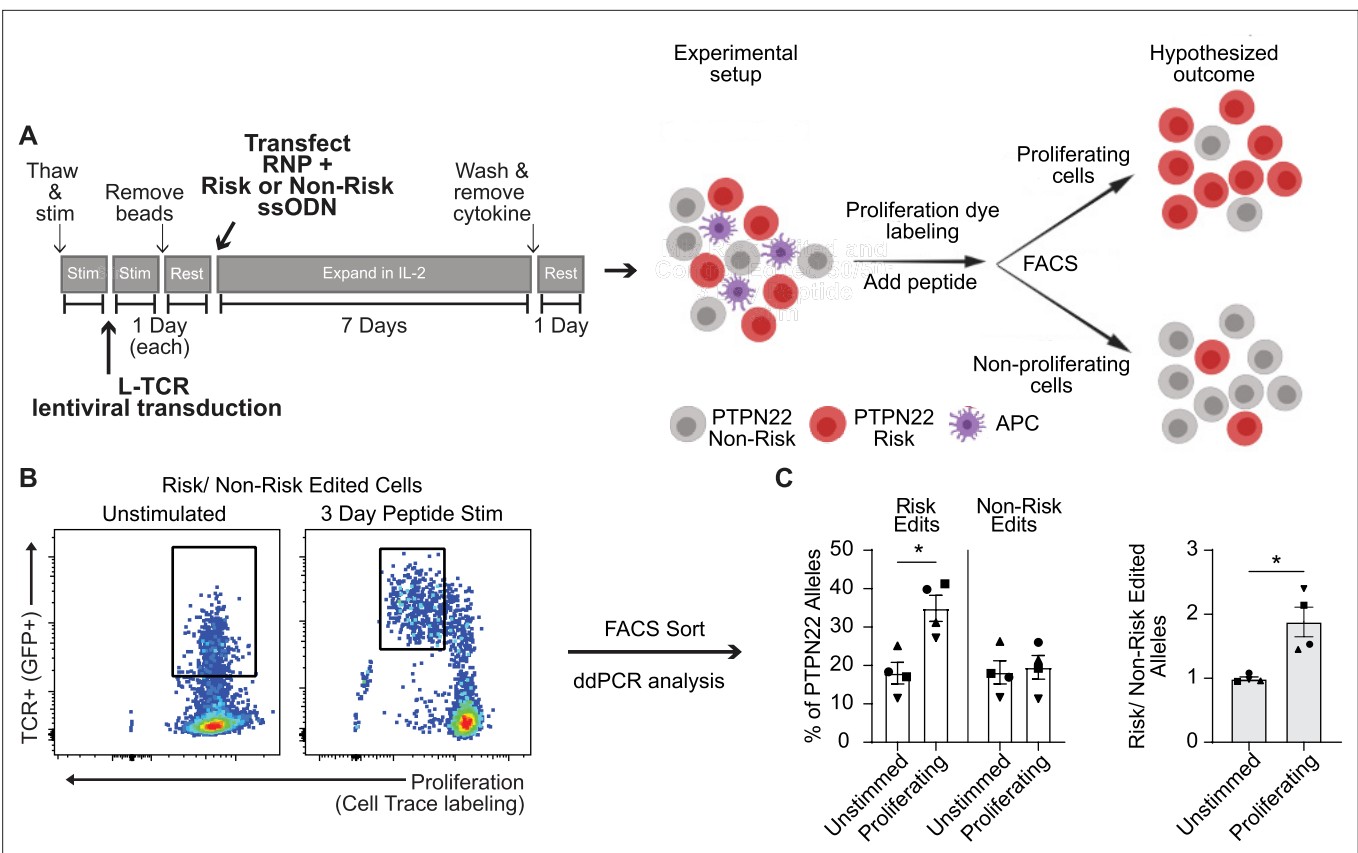

**Figure 6.** Protein tyrosine phosphatase non-receptor 22 (*PTPN22*) risk edited T cells show increased proliferation compared to non-risk edited cells under L-TCR stimulation. Cord blood CD4 T cells from human donors were transduced with L-TCR lentivirus prior to gene editing into two populations (as described in *Figure 1*): *PTPN22* risk edited, or non-risk edited, then stimulated with cognate peptide. (**A**) Workflow shown in *Figure 3*, depicting the use of *PTPN22* single nucleotide polymorphism (SNP) locus editing DNA repair templates and an experimental model describing how the *PTPN22* R620W coding sequence's impact on proliferation was predicted and would be assessed. (**B**) A representative FACS plot showing gating of L-TCR+ populations before peptide stimulation and 3 days post peptide stim. (**C**) Droplet digital PCR (ddPCR) analysis of edited *PTPN22* alleles in FACS sorted populations. n=4 independent human donors (shapes correspond to individual donors). All summary data analyzed by paired t test. *p<0.05.

The online version of this article includes the following source data and figure supplement(s) for figure 6:

**Source data 1.** Competitive proliferation, protein tyrosine phosphatase non-receptor 22 (PTPN22) edited cells.

**Figure supplement 1.** The proportion of protein tyrosine phosphatase non-receptor 22 (*PTPN22*) risk edited T cells is also increased within the non-proliferating cell population following L-TCR engagement.

**Figure supplement 1—source data 1.** Protein tyrosine phosphatase non-receptor 22 (PTPN22) risk variant increases numbers of non-proliferating cells.

Because *PTPN22* risk variant edited T cells exhibited a phenotype resembling *PTPN22* KO T cells in response to anti-CD3 stimulation, we next determined whether risk variant edited cells exhibited increased proliferation in response to cognate antigen. Generation of *PTPN22* risk edited cells leads to a reduction in *PTPN22* expression due to competing HDR and NHEJ events (*Figure 1*). Thus, to control for levels of HDR edited vs. gene disrupted *PTPN22* alleles, we designed our experiments to assess the functional outcome of TCR engagement within a pool of risk vs. non-risk edited cells. To achieve this comparison, cord blood T cells were LV transduced to deliver the L-TCR and subsequently edited (as in *Figure 1A* and *Figure 3A*) to generate *PTPN22* risk edited or non-risk edited popula-tions, each expressing L-TCR (*Figure 6A*). Next, edited populations were mixed in equal proportions, and labeled with a proliferation dye. This mixed population was then used to test the hypothesis that, in response to stimulation with cognate antigen, *PTPN22* risk edited cells would become the domi-nant genotype within the proliferating cell population. gDNA based ddPCR analysis on FAC sorted TCR+ cells prior to stimulation demonstrated equal proportions of *PTPN22* risk and non-risk edited alleles (*Figure 6B–C*). The mixed population was stimulated using peptide loaded APCs for 3 days and proliferating vs. non-proliferating cells were isolated by FAC sorting (*Figure 6B*; *Figure 6—figure*

*supplement 1A–B*). ddPCR revealed nearly twice as many *PTPN22* risk edited over non-risk edited alleles within the proliferating cell population (*Figure 6B–C*). Together, this data supports a hypothesized model (*Figure 6A*) wherein human T cells expressing the *PTPN22* risk variant exhibit increased proliferation in response to low-avidity TCR engagement. Interestingly, gDNA taken from stimulated but non-proliferating TCR+ cells also showed a partial increase in *PTPN22* risk edited alleles relative to input. While this difference was less robust than among proliferating cells (*Figure 6—figure supplement 1A–B*), it suggests that increased TCR signal strength in risk variant T cells may promote enhanced cell survival, a finding that will require additional study.

## Discussion

In this paper we utilized a robust gene editing platform in primary human cord blood CD4 T cells to efficiently alter *PTPN22* non-risk T cells to express the *PTPN22* R620W risk variant as well as relevant control, non-risk, or KO edited cell populations. This gene editing approach should be applicable to many cell types and broadly applicable to study the impact of candidate disease associated alleles (identified by GWAS or other sequencing strategies) within primary human cell populations. Notably, our data clearly demonstrate that *PTPN22* risk variant expressing T cells exhibit a phenotype similar to *PTPN22* KO T cells, leading to increased expression of TCR induced activation responses markers. Further, using transgenic TCR delivery and cognate peptide-mediated activation, in tandem with our gene editing platform, we show that PTPN22 deficiency predominantly impacts T cells activated in response to low-avidity TCR engagement. In contrast, PTPN22 function is largely dispensable in T cells receiving a high-avidity TCR signal. Together, these data suggest that the *PTPN22* risk variant may contribute to autoimmune responses, at least in part, by selectively expanding the pool of lower avidity, self-reactive T cells and skewing such cells toward a pro-inflammatory phenotype. Our findings are consistent with previous work in a fixed TCR murine model that also found loss of ptpn22 to preferentially impact low-avidity TCR stimulation (*Salmond et al., 2014*). Importantly, our study models function of the PTPN22 regulatory pathway in a fully human gene edited primary cell setting.

Our editing platform utilizes a designer nuclease that generates a dsDNA break in close proximity to the target site for HDR editing. This combined approach allows for: (a) generation of edited cells with high rates of HDR; (b) equivalent HDR rates for alternative coding changes within the target gene via use of a series of ssODN donors; and (c) achieving editing outcomes while controlling for the overall HDR/NHEJ ratio. An additional strength is that the location and small editing footprint allows for physiological expression via use of the endogenous promoter, local enhancers, and 3' UTR. Also, editing in this location is unlikely to disrupt the formation of natural *PTPN22* splice variants. Of note, *PTPN22* risk and non-risk ssODNs, while producing identical HDR rates, demonstrated reduced HDR relative to editing using the *PTPN22* KO ssODN. This modest difference likely reflects the requirement to replace endogenous nucleotides immediately 5' and 3' to the DNA cut site to achieve risk or non-risk edits, while the KO ssODN only required insertion of new nucleotides. These minor differences aside, our editing platform should be readily adaptable for the study of multiple loci across the genome in future studies of human primary cell populations.

We demonstrate that expression of the *PTPN22* risk variant can increase proliferation of primary human T cells expressing a low-avidity TCR, findings that mimic *PTPN22* KO T cells. Furthermore, we show that loss of PTPN22 regulation promotes the development of a pro-inflammatory phenotype in T cells using a low-avidity TCR. These combined findings are consistent with in vivo functional impacts observed in human and mouse studies. Indeed, while previous human data and mouse models have led to contradictory conclusions regarding the impact of the risk variant on proximal TCR signaling, both humans and mice with the *PTPN22* risk variant exhibit expanded memory T cell and Th1 populations (*Rieck et al., 2007*; *Vang et al., 2013*; *Dai et al., 2013*; *Salmond et al., 2014*). Importantly, *PTPN22* is expressed in multiple immune and hematopoietic subsets and has been shown to have multiple functions (*Rawlings et al., 2015*; *Stanford and Bottini, 2014*; *Armitage et al., 2021*). Therefore, our dataset informs on a single signaling program and *PTPN22* may contribute to autoimmune disease via other pathways.

While our editing design provides a useful means to address questions related to TCR engagement, this approach has limitations including assessing early intracellular signaling events. While gene edited cells exhibited proximal phosphorylation changes in response to antibody-based stimulation, biochemical changes were not detected until 24 hr post peptide-based stimulation. Future work may

find a path around this technical challenge allowing better insight into early signaling events. Also, while our findings provide an important bridge between inconsistencies connecting *PTPN22* to auto-immune risk in previous mouse vs. primary human cell data, it should not be viewed as a replacement for mouse models. Our study relies on ex vivo culturing of T cells, outside of optimal physiological conditions and supportive niches, features that may have key impacts on protein tyrosine phosphatase functions. Also, though our gene KO model reached 100% as measured by ddPCR, western quantifi-cation data (*Figure 1E*) may suggest residual expression of *PTPN22*, a consideration not of relevant for mouse models. Finally, as our model is restricted to peripheral mature T cells, it is inadequate to assess the role for PTPN22 in T cell development. For example, while mouse models suggest that the *PTPN22* risk variant alters thymic selection (*Dai et al., 2013*) others have shown that *PTPN22* KO has no impact on the TCR repertoires (*Hasegawa et al., 2004*; *Salmond et al., 2014*; *Sanchez-Blanco et al., 2018*; *Maine et al., 2012*). Because our study shows the risk variant functions similarly in human T cells in comparison with published murine models, it suggests that returning to these models to investigate specific contexts where PTPN22 might impact selection and tolerance is likely to be infor-mative and critical for full understanding of this variant.

While T cells expressing a lower avidity TCR were primarily impacted in our studies, we cannot eliminate potential alternative impacts of the TCRs studied in our models. The TCRs evaluated exhib-ited distinct differences in both $EC_{50}$ (*Linsley et al., 2021*) and triggered proliferative response to an identical IGRP derived peptide (*Figure 3C*; *Linsley et al., 2021*). However, while these data points support a working hypothesis of TCR avidity determining the functional impact, we cannot rule out the role for possible differences in TCR disassociation rates and/or overall duration of TCR engage-ment. Further, while our approach provides a robust system to assess the impact of the risk variant in primary human T cells, our findings are derived from a comparison of only two specific TCRs. The identification of similar pairs of 'higher' and 'lower' avidity TCRs recognizing identical self-peptides will be useful in assessing our model and its interpretation across a broader pool of TCR interac-tions. Notably, in this study we explored the impacts of the *PTPN22* risk variant and KO specifically in cord blood CD4 T cells. CD4 T cells were chosen for this study based on well-documented use in gene editing models and likely key roles in *PTPN22* associated autoimmune diseases (*Zheng et al., 2012*). Cord blood was used to minimize differences in environmental impact between T cell donors. Although previous work has revealed similar findings regarding the phenotype of *PTPN22* KO in adult T cells (*Anderson et al., 2019*), it remains possible that similar risk variant editing experiments using adult peripheral blood may yield different results.

Consistent with our findings, recent work using lentiviral-based overexpression in primary human T cells also revealed hypomorphic function of the *PTPN22* risk variant (*Perry et al., 2021*). Further, that study and others *Sanchez-Blanco et al., 2018*; *Burn et al., 2016* have suggested that PTPN22 may modulate pathways other than the TCR including LFA-1-mediated integrin signals, stabilizing immunological synapses that promote T cell proliferation and Th1 skewing. Our data using low dose CD3 stimulation (*Figure 2*) support the conclusion that both the variant and *PTPN22* KO can impact T cell activation through altered regulation of the TCR. Notably, our CD3 studies were performed in the absence of co-stimulation to better mimic sub-optimal TCR engagement. Thus, the relevance of our findings based on endogenous-level expression compared to lentiviral-based *PTPN22* over-expression in association with CD3/CD28 co-stimulation are difficult to compare. Together, our find-ings that the risk variant preferentially modulates low avidity TCR engagement, with the observations of these previous studies suggests that the *PTPN22* risk variant may exert its impact on T cells both via altering TCR signaling and by modulating additional signals including LFA-1 pathway regulation.

Through gene editing we demonstrate that *PTPN22* risk variant expression in human T cells medi-ates at least a partial loss of function phenotype that mirrors findings derived from mouse models. This data, however, does not explain the marked differences in T cell responses described in mouse models vs. natural carriers of the *PTPN22* risk variant. Natural carriers of the risk variant exhibit hypore-sponsive T cells, with reduced TCR induced activation, and reduced cytokine responses (*Rieck et al., 2007*; *Vang et al., 2005*). In previous work, we showed that loss of function in another key nega-tive regularity pathway in T cells leads to overexpression of functionally related proteins to maintain homeostasis. For example, KO of the IL-2 regulatory gene, *PTPN2*, in primary human T cells results in a transient increase in IL-2 signaling, but expansion of PTPN2 KO cells in high cytokine media leads to reduced IL-2 responses due to overexpression of the regulatory protein SOCS3 (*Anderson et al.,*

*2019*). Similar compensatory events may be operable in T cells in carriers of the *PTPN22* risk variant, where reduced proximal negative regulatory events might lead to expression of alternative regulatory proteins. Recent work using optogenically controlled TCR signaling has shown that T cell activation is dynamic, requiring optimal frequencies of TCR engagement relative to cessation of signal to mediate CD69 expression and subsequent activation; however, events that limit T cell responses may be more static, with expression of TIM3, for example, increasing regardless of signaling frequency (*O'Donoghue et al., 2021*). As loss of PTPN22 function specifically augments low-avidity TCR signaling, it is possible that tonic TCR signaling is also impacted and may drive additional events that function to limit TCR signals in carriers of the risk variant over time. Finally, it is important to highlight the environmental differences between murine models and primary human T cell donors. As opportunity for exposure to antigen increases, this may drive T cells from human carriers toward a reduced responsiveness to TCR engagement via impacts that are not operative in SPF murine models.

In summary, the data presented within this study supports a model where the *PTPN22* R620W risk variant impacts human T cell function via a reduction in negative regulation of TCR signaling; findings similar to murine models, but in contrast to the T cell signaling phenotype in carriers with the risk variant. In association with previous human studies, our findings suggest that the impact of the risk variant on T cell function is likely dynamic, leading to the generation of populations of hyperresponsive T cells, but also allowing for secondary impacts on T cell gene expression that may lead to hyporesponsive T cells over time or after environmental exposures. This would help explain how *PTPN22* could influence risk of diverse autoimmune diseases such as type 1 diabetes and rheumatoid arthritis, diseases with distinct ages of onset and tissue targets. Taken together, these observations suggest that *PTPN22* contributes to autoimmunity by establishing temporal windows of risk for specific diseases that coincide with periods of altered T cell responsiveness.

## Materials and methods

### Primary T cell isolation and culturing

PBMCs were collected from whole (peripheral or cord) blood of consenting donors and cryopreserved. Upon thaw, naive CD4$^+$ T cells were isolated by negative selection (Naïve CD4 T Cell Isolation Kit II, Miltenyi Biotech) and cultured in Roswell Park Memorial Institute (RPMI) 1640 media supplemented with 20% FBS, 1× Glutamax (Gibco), and 1 mM HEPES (Gibco). Unless otherwise noted, cells were cultured in 5 ng/ml recombinant IL-2 (Peprotech). After thaw, cells were counted and cultured at 1 million/ml in flat bottom culture plates.

### CRISPR/Cas9 and ssODN reagents

CRISPR gRNAs targeting *PTPN22* exon 14 were commercially synthesized by Synthego (CRISPRevolution sgRNA). ssODNs were commercially synthesized by Integrated DNA Technologies (IDT; Ultramer DNA Oligonucleotides) with phosphorothioate linkages between the first and final three base pair sequences. gRNA complexes were mixed with Cas9 nuclease (IDT) at a 1.2:1 ratio and delivered with or without ssODNs into cells by electroporation. *PTPN22* targeting 20 bp gRNA sequence used for all editing: AATGATTCAGGTGTCCGTAC.

### Antigen specific TCR LV

The islet antigen-specific TCRs utilized in this study have been previously described (*Cerosaletti et al., 2017*; *Linsley et al., 2021*). These TCRs each recognize the identical peptide derived from the islet autoantigen, IGRP. T1D2 TCR herein described as L-TCR exhibits an EC50 of 0.97 µg/ml and T1D5-2 TCR herein described as H-TCR, exhibits an EC50 0.09 µg/ml. TCR coding sequences were cloned into lentiviral backbones using InFusion HD (Takara), with oligonucleotides synthesized by IDT, and a cis-linked (T2A) eGFP was inserted into the 3' end of the alpha constant region. LV-TCR plasmids were based on the pRRL backbone derived from pRRLSIN.cPPT.PGK-GFP.WPRE (plasmid #12252, Addgene). The DNA sequences of all plasmids were verified by Sanger sequencing performed by GENEWIZ. Methods for LV production have been previously described (*Seymour et al., 2021*). Lentivirus stocks were tittered by rtPCR.

## Gene editing

After thaw, cells were activated with CD3/CD28 Activator Beads (Gibco). After 2 days, beads were magnetically removed and cells re-plated without changing media or adjusting cell number. One day later, cells were electroporated with editing reagents.

Prior to electroporation, cells were washed with PBS and resuspended in Lonza P3 buffer. Five µg of complexed RNP and 100 pmol ssODN per $1×10^6$ cells was added to the resuspension so that the final cell density was $5×10^7$ cells/ml. Cells were electroporated with a Lonza 4-D Nucleofector, using program DN-102 with 16-well 20 µl Nucleocuvette strips or 100 µl Nucleocuvette vessels, and then transferred into pre-warmed cell culture medium with IL-2 (unless otherwise noted). For samples transduced with lentivirus, virus was added to the culture 1 day after bead stim addition at an MOI of 5. After editing, cells were maintained in media identical to pre-editing conditions (unless otherwise noted). Cells were counted at least every 2 days using Count Bright absolute counting beads (Thermo Fisher Scientific) and split to maintain cell density of 1–2 million/ml. Following expansion cells were counted, washed two times with PBS, and rested at 1.5 million/ml for 24 hr in cytokine free media consisting of RPMI 1640 media with 10% FBS, 1× Glutamax (Thermo Fisher Scientific), and 1 mM HEPES. Cells were re-counted prior to stimulation. For some experiments, cells were frozen after 24 hr cytokine free rest, then thawed/counted prior to stimulation.

## ddPCR

Quantification of HDR and NHEJ rates in edited human CD4+ T cells was obtained using a ddPCR, dual-probe competition assay. All probes were ordered from Sigma-Aldrich with a 3′ Black Hole 1 Quencher. Probes specific to sequences generated by HDR insertion of SNP edits or stop codons were labeled with a 5′ FAM reporter and used in tandem with a 5′ HEX labeled probe specific to wild type (WT) sequences. Editing was measured after generating droplets with 50 ng of genomic DNA (gDNA), both HDR-FAM and WT-HEX probes, and primers to the editing locus producing amplicons of <500 bp (1× assay, 900 nM primers, and 250 nM probe) using ddPCR supermix for probes (no deoxyuridine triphosphate) (Bio-Rad). Reference reactions were simultaneously performed using a 5′ HEX labeled probe/primer combination targeting the housekeeping gene B2M. Droplets were generated with the QX200 Droplet Generator (Bio-Rad) and amplified. All samples were run in triplicate and averaged. Fluorescence was measured using the QX200 Droplet reader (Bio-Rad) and analyzed using Quantasoft software. Editing rates were calculated as the relative frequency (%) of FAM+ corresponding to %HDR, HEX+ corresponding to %No Event, and reference – (FAM+HEX) corresponding to %NHEJ.

For ddPCR analysis of co-cultured *PTPN22* risk/non-risk edited cells HDR (risk) -FAM and HDR (non-risk) -HEX probes with locked nucleic acid (LNA) modifications were utilized. LNA modified nucleotides were integrated at the SNP nucleotide and the adjacent 5′ and 3′ nucleotides to increased probe discrimination. Risk/non-risk probe reads were compared to a B2M reference reaction as above.

## Western blotting

All western blots were performed on lysates from human umbilical cord blood-derived CD4 T cells that had been either mock edited or gene edited, expanded 7 days in cytokine supplemented media, and subjected to 24 hr rest in cytokine free media. Cells were lysed in 1× RIPA lysis buffer on ice for 10 min then clarified by centrifugation. Concentration of clarified lysate was determined by BCA assay (Pierce), diluted, and suspended in 1× LDS Sample Buffer (Invitrogen). Ten µg of lysate was run on 4–12% Bis-Tris NuPAGE gels in 1× MOPS buffer (Invitrogen). Protein was transferred to nitrocellulose in 1× Transfer Buffer (Invitrogen) and 10% methanol. Non-specific binding was minimized with a 1 hr room temperature (RT) incubation in Odyssey LI-COR Blocking Buffer. Primary antibodies were stained at 1:1000. PTPN22 primary stain was for at least 12 hr at 4°C and actin was stained at RT for 40 min. Primary antibodies used were from Cell Signaling Technology: PTPN22 (D6D1H, Cat# 3700, RRID:AB_2798575) and actin (8H10D10, Cat# 14693, RRID:AB_2242334). After primary stain membranes were washed with 1× TBST and incubated with secondary antibodies at 1:10,000 for 30 min at RT. Stained blots were washed and imaged on an Odyssey Infrared Imaging System (LI-COR Biotech.). Western blot quantifications were performed with ImageJ software.

## Plate bound anti-CD3 stimulation and Th1 skewing

Stimulation plates were made in 96-well flat bottom culture plate. One-hundred μl of PBS supplemented with LEAF purified anti-CD3 (OKT3, Biolegend, Cat# 317326, RRID:AB_11150592) at 0.25 μg/ml was added to each well and incubated at least 12 hr at 4°C. The plate was then emptied, and wells were given 100 μl of cytokine free T cell media. After cells were edited, expanded for 7 days, and rested 24 hr in cytokine free media, 100 μl of cells at 2 million/ml were added to each well. Plates were incubated at 37°C for up to 48 hr.

For Th1 skewing of edited T cells, cells that had been edited and rested were transferred to 96-well flat bottom culture plates, coated with anti-CD3 as discussed above. These cells were cultured for 3 days in media containing 20 ng/ml IL-12 (Peprotech), 50 ng/ml IL-2, 40 μg/ml anti-IL-4 (MP4-25D2, eBioscience Cat# 14-7048-81, RRID:AB_468414), and 500 ng/ml soluble anti-CD28 (CD28.2, BioXcell Cat# BE0291, RRID:AB_2687814) to promote a Th1 phenotype. Non-skewed controls were cultured in media that contained IL-2, anti-CD3, and anti-CD28 only.

## Peptide stimulation of antigen specific edited T cells

LV-TCRs used in this study are HLA-DRB1 04:01 restricted and recognize the 20-mer peptide sequence: QLYHFLQIPTHEEHLFYVLS (IGRP- p39 position 305–324). For stimulation of cells expressing LV-TCRs, antigen presenting cells (APCs) were PBMCs from an allogeneic HLA-DRB1 04:01 donor irradiated with 5000 rads. APCs were then mixed with peptide diluted in complete media (final peptide concentration 50 ng/ml) for 2 hr. Co-cultures were incubated in 96-well U-bottom plates with 200 K APCs and 50 K T cells per well at 37°C for up to 3 days in complete media containing no exogenous cytokines. APCs for all experiments were from the same PBMC donor.

## Flow cytometry and gating strategies

Flow cytometric analysis was performed on an LSR II flow cytometer (BD Biosciences) and data was analyzed using FlowJo software (Tree Star). Cells were stained with LIVE/DEAD Fixable Near-IR Dead Cell Stain Kit (Fisher Sci), as per the manufacturer's instructions and cells were stained with fluorescence labeled antibodies for 30 min on ice. In this study, flow cytometry antibodies used are from Biolegend: CD3 (UCHT1, Cat# 300434, RRID:AB_10962690), CD4 (RPA-T4, Cat# 300556, RRID:AB_2564391), CD69 (FN50, Cat# 310910, RRID:AB_314845), CD25 (M-A251, Cat# 356112, RRID:AB_2561979), CD71 (CY1G4, Cat# 334112, RRID:AB_2563119), IFNγ (4S.B3, Cat# 502526, RRID:AB_961355), and anti-mouse TCR-β (H57-597, Cat# 109212, RRID:AB_1 313435), from Thermo Fisher Scientific: IL-2 (MQ1-17H12, Cat# 25-7029-42, RRID:AB_2573518), and from Cell Signaling: phospho-S6 kinase (D57.2.2E, Cat# 5316, RRID:AB_10694989).

All surface antibodies were used at a dilution of 1:100, antibodies against cytokines were used at a dilution of 1:200, and those staining phospho-sites were used at a dilution of 1:50. Gating order proceeded: lymphocytes→ singlets→live cells. Surface stains of other markers were subsequently gated on CD4+/TCR+ cells, then the marker of interest. For pS6 staining, cells were fixed with a final concentration of 2% PFA for 12 min at 37°C. Cells were then washed and permeabilized with BD Perm Buffer III for at least 30 min at –20°C. Cells were then washed and stained as described above. For cytokine staining, after 3 days of stimulation cells were stimulated with 50 ng/ml PMA, 1 μg/ml ionomycin, and 1 μg/ml monensin for 5 hr, then fixed/permed with BD Cytofix/Cytoperm prior to staining for cytokine production. For assessment of proliferation, peptide stimulated antigen specific T cells were labeled with CellTrace Violet proliferation dye (Fisher Science) prior to co-culturing with APCs; number of antigen specific T cells in each edited sample was determined daily by addition of Countbright Absolute Counting Beads (Fisher Science) in flow cytometry samples.

## Experimental design

All cells used in this study are primary human cells. With the exception of *Figure 1—figure supplement 1B* all data depicts experiments performed using umbilical cord blood-derived T cells, isolated within 36 hr of birth. Sex of donor was not held constant, and all figures reflect data generated with male and female derived materials, and ethnicity data was not collected. Once blood was confirmed to be homozygous for the non-risk (620R) PTPN22 variant to allow efficient gene editing, no exclusion criteria were applied. All experiments were performed two to four times as independent experiments (indicated in each figure legend), and N is indicated in each figure legend, and as each data point

represents an independent human sample all data points meet the definition of 'biological replicates', with the exception of *Figure 1—figure supplement 1B*, which was performed in a single experiment with two independent human donors and three technical replicates per donor.

## Statistics

Statistical analyses were performed using GraphPad Prism 9 (GraphPad). For all testing of gene edited cells, due to the low variability in culturing conditions and lack of obvious skewing, data was assumed to maintain a normal distribution. p-Values in multiple comparisons were calculated using one-way ANOVA with the Tukey's correction; p-values in comparisons between two groups were calculated using a paired two-tailed t test. Values from combined independent experiments are shown as mean ± SEM.

## Study approval (human subjects)

For gene editing experiments using adult PBMCs, human donor leukopacks were purchased from the Fred Hutchinson Cancer Research Center, which were obtained from consenting donors under an IRB-approved protocol and cryopreserved. For gene editing experiments using umbilical cord blood-derived PBMCs, cord units were purchased from the Bloodworks Northwest, which were obtained with consent under an IRB-approved protocol and cryopreserved. After collection, all samples were de-identified for the protection of human blood donors.

## Acknowledgements

We thank Rich James (Seattle Children's Research Institute, Seattle, WA) for his helpful comments and insight.

## Additional information

### Funding

| Funder | Grant reference number | Author |
| --- | --- | --- |
| National Institute of Diabetes and Digestive and Kidney Diseases | DP3DK111802 | Warren Anderson<br>Fariba Barahmand-pour-Whitman<br>Peter S Linsley<br>Karen Cerosaletti<br>Jane H Buckner<br>David J Rawlings |
| Seattle Children's Foundation | Children's Guild Association Endowed Chair in Pediatric Immunology | David J Rawlings |
| Seattle Children's Research Institute | Center for Immunity and Immunotherapies | David J Rawlings |
| Seattle Children's Research Institute | Program for Cell and Gene Therapy | David J Rawlings |
| Seattle Children's Research Institute | Hansen Investigator in Pediatric Innovation Endowment | David J Rawlings |
| Benaroya Family Gift Fund | | David J Rawlings |

The funders had no role in study design, data collection and interpretation, or the decision to submit the work for publication.

### Author contributions

Warren Anderson, Conceptualization, Data curation, Formal analysis, Validation, Investigation, Methodology, Writing - original draft, Writing - review and editing; Fariba Barahmand-pour-Whitman, Conceptualization, Resources, Investigation, Methodology, Writing - review and editing; Peter S

Linsley, Karen Cerosaletti, Jane H Buckner, Conceptualization, Resources, Methodology, Writing - review and editing; David J Rawlings, Conceptualization, Resources, Supervision, Funding acquisition, Methodology, Project administration, Writing - review and editing

### Author ORCIDs
Warren Anderson  http://orcid.org/0000-0002-9315-5233
Peter S Linsley  http://orcid.org/0000-0002-8960-4307
Karen Cerosaletti  http://orcid.org/0000-0002-7403-6239
David J Rawlings  http://orcid.org/0000-0002-0810-1776

### Ethics
Study Approval (Human Subjects)For gene editing experiments using adult PBMCs, human donor leukopaks were purchased from the Fred Hutchinson Cancer Research Center, which were obtained from consenting donors under an IRB-approved protocol and cryopreserved. For gene editing experiments using umbilical cord blood derived PBMCs, cord units were purchased from the Bloodworks Northwest, which were obtained with consent under an IRB-approved protocol and cryopreserved. After collection, all samples were de-identified for the protection of human blood donors.

### Decision letter and Author response
Decision letter https://doi.org/10.7554/eLife.81577.sa1
Author response https://doi.org/10.7554/eLife.81577.sa2

---

## Additional files

### Supplementary files
- MDAR checklist
- Source data 1. Zip file of all western blot pictures, raw and annotated.
- Source data 2. Zip file of all source data for summary data graphs.
- Source data 3. Short DNA sequences.

### Data availability
Prism files containing the numerical data and statistical tests used to generate all figures has been provided.

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
