## [Editor Report]

Although the experimental system is limiting in addressing intracellular signaling parameters, the overall conclusions of the paper are important in that they further our understanding of the mechanisms through which the PTPN22 R620W variant, associated with multiple autoimmune diseases, contributes to breech of peripheral T cell tolerance. Moreover, this work greatly advances and clarifies ongoing confusion about whether PTPN22 SNP(620W) is a Loss Of Function mutant.

---

## [Decision Letter]

**Decision letter after peer review:**

Thank you for submitting your article "PTPN22 R620W gene editing in T cells enhances low avidity TCR responses" for consideration by *eLife*. Your article has been reviewed by 3 peer reviewers, and the evaluation has been overseen by a Reviewing Editor and Tadatsugu Taniguchi as the Senior Editor. The following individuals involved in review of your submission have agreed to reveal their identity: Andrew Cope (Reviewer #1); Andrew C. Chan (Reviewer #3).

Essential revisions:

Two issues need to be addressed. First, some additional assessment of intracellular signaling parameters to distinguish between total or hypomorphic LOF. Second, as Reviewers 1 and 3 note, ligand titration of the high-affinity TCR is needed to better support the lack of phenotype for PTPN22 loss for high-afifnity interactions.

*Reviewer #1 (Recommendations for the authors):*

This reviewer commends the authors on the near Hereculean efforts undertaken to generate the range of genetically modified cord blood T cells for study.

General comments:

It might be worth mentioning the specificity of the TCRs in the abstract just to give the story context. It took this reviewer a while to find this info, buried in methods. The Results section reads a bit like a discussion or even review paper, focused on the nuances of gene editing. This reviewer would suggest keeping the narrative of the results focused on describing the experimental findings, and confining technological limitations and challenges to the discussion. This is a somewhat stylistic issue, but would make for a better read.

*Reviewer #2 (Recommendations for the authors):*

Overall, this conclusion is well funded based on the experiments shown, with only a few issues or questions (see below).

1) In the figures the different symbols represents different donors, which is clearly marked. However, the data in all experiments are from different experiments, it would be helpful if it could be marked also which symbols belong to which experiment (maybe using different colors) to get a grip on the interexperimental variation ( I understood the values are not normalized?).

2) The reference list is dominated by reviews. It is important to refer to original publications not only because it is more fair but also to prevent false statements going into circuits. Pls take away most, or preferably all, of these references and replace with original when needed.

3) In the discussion regarding different outcomes in mouse versus human it is pointed out that the mouse is in SPF conditions and are genetically more homogenous. This is true but there are likely even more important issues regarding experimental settings. Just to mention taking out T cells from their normal biological context, put them in an incubator with atmospheric oxygen pressure represent a shock and dramatically changes the context. This is of particular importance for PTPs, including PTPN22, which are profoundly redox regulated. There are numerous differences and usually the most important differences are the experimental settings and I think it should be pointed out and discussed in a better way. For future studies, the mouse is absolutely needed, for example, the lymphocyte selection, both T and B, needs to be addressed and I guess experimental mice are critically needed to solve this. Thus, the main conclusion from this paper is that it provides good arguments to improve the animal models to solve the functional genetics of PTPN22 (and most likely other SNPs).

*Reviewer #3 (Recommendations for the authors):*

This work employs an experimental system that minimizes human donor heterogeneity and developmental skewing. The authors demonstrate that PTPN22 plays an inhibitory role on TCR regulated effector functions in low, but not high, avidity TCR interactions. The work builds on a report by Salmond et al. (Nature Immunology, 15:875, 2014) using the murine OT-I TCR system that PTPN22 restrains TCR signaling induced by lower affinity agonists (T4 or G4 peptides) than the higher affinity N4 agonist. In the present work, the authors demonstrate further in human T cells that PTPN22 KO or SNP edited cells demonstrate similar enhanced TCR-mediated functions when compared to PTPN22 wildtype T cells. The authors should consider the following:

1. As the highest peptide dose was utilized in the signaling studies in Figures 4-6 since this dose produced efficient proliferation, the authors should consider performing a dose-titration with the H-TCR studies to fully support their conclusion that there are no differences in H-TCR signaling in the presence or absence of PTPN22.

2. Page 19, line 439- the level of TCR expression in the transduced cells should be shown.

3. The conclusions of the Salmond et al. study mentioned above describing differences in low vs high affinity agonists should be provided greater description as background.

4. The authors should also discuss how their conclusions compare with the recent study of Perry et al. (Journal of Immunology, 207:849, 2021) where using an overexpression system, they conclude that the PTPN22(620W) SNP is hypomorphic in human CD4^+^ T cells rather than a full loss-of-function.

---

## [Author Response]

Essential revisions:Two issues need to be addressed. First, some additional assessment of intracellular signaling parameters to distinguish between total or hypomorphic LOF. Second, as Reviewers 1 and 3 note, ligand titration of the high-affinity TCR is needed to better support the lack of phenotype for PTPN22 loss for high-afifnity interactions.

(1.) Below we discuss our attempts to assess intracellular signaling in T cells with total or hypomorphic loss of PTPN22 function.

We agree that differences in intracellular signaling events between *PTPN22* WT and *PTPN22* LOF cells using our TCR/antigen specific model would be useful and informative. However, use of antigen presenting cells for peptide stimulation demonstrated a delayed and muted phosphorylation response in comparison with antibody-based activation (likely due to differences in TCR/ MCH binding kinetics relative to that of anti-CD3, with or without anti-CD28 antibodies).

We have tried to investigate the pathways suggested by the reviewers (pErk and pAKT) with limited success. Importantly, we argue that the key finding in our study reflects the critical role for PTPN22 in low avidity (or perhaps low antigen density) TCR signaling in naïve T cells. Thus, we focused on signaling assays using peptide loaded APCs as a stimulant. We first attempted to find ideal timepoints in stimulation that would accentuate any differences in the kinetics of these pathways between *PTPN22* edited cells.

Early time course testing using CD3/CD28 beads on gene edited /TCR+ T cells suggested that optimal timing for measuring differences in pErk occurred between 15-30 minutes post activation when using CD3/28 crosslinking. See Author response image 1.

**Author response image 1. sa2fig1:** 

However, upon switching to stimulation with APCs (with or without cognate peptides) and using this kinetic, we found no differences in pERK in the presence or absence of peptide. See Author response image 2.

As we did not observe significant increases in pErk over a 1-hour time course in the setting of peptide loaded APCs, we shifted our time course to hours (presuming the signaling kinetic was delayed relative with CD3/28 simulation beads). However, repeating these experiments over a 6 hr time course, addition of peptide to the APCs did not lead to increased pErk relative to cells stimulated with APCs alone. Please see Author response image 3. Consistent with failure to observe pERK activation, similar inconclusive data was generated using pAKT and total pTyr intracellular staining (not shown).

**Author response image 3. sa2fig3:** 

We next assessed a longer stimulation timeline (24 hrs). In that setting, we observed increased pErk in response to peptide stimulation in T cells expressing the H-TCR, but not with L-TCR transduced cells. However, in keeping with our findings described in the manuscript, we observed no differences between H-TCR stimulated cells, with respect to PTPN22 status. See Author response image 4.

**Author response image 4. sa2fig4:** 

As use of peptide loaded APCs limited our ability to assess proximal signaling in our model, we next elected to investigate the impact of the PTPN22 on pS6 and p4-EBP1. At a 24-hour timepoint, we observed a modest increase in pS6 in PTPN22 KO cells transduced with L-TCR relative to PTPN22 WT cells and a similar trend in p4-EBP1. Incorporating PTPN22 non-risk and PTPN22 risk edited cells in these assays produced responses similar to activation marker data shown in Figure 2 in our study. However, these data did not reach statistical significance for the risk variant allele. Based on the Reviewer and Editor’s comments, we have added these pS6 and p4ebp1 data to the revised manuscript as Figure 3 —figure supplement 2.Finally, as we have been unable to adequately assess differences in early intracellular signaling pathways, we agree that this represents a weakness of this model. We have revised the Discussion section to point out this limitation of our work (Lines: 440-444).

(2.) Response to the comment that additional ligand titration of the high-affinity TCR is needed to better support the lack of phenotype for PTPN22 loss in high-affinity interactions.

We appreciate the request to include more data to show a lack of phenotype between PTPN22 WT and KO in H-TCR stimulated cells across a range of peptide concentrations. We previously generated these data to convince ourselves that the PTPN22 LOF phenotype was only evident under L-TCR stimulation and did not just represent a missing optimized condition for assessing H-TCR stimulation. See Figure 3 and Figure 3 —figure supplement 2.

We had chosen to streamline the figure in the initial manuscript for ease of communication. However, we agree that this data is informative in verifying our conclusions, and we have now included it in the final manuscript. We used “XY” graph to replace the previous plot in Figure 3C and relabeled the new plot figure 3D and we have included the t tests of the area under the curve (AUC) in new Figure 3 —figure supplement 2.

Of note, we utilized AUC in lieu of ANOVA to analyze differences between peptide dose titrations as we found it to be the better test for communicating our findings. Using ANOVA to analyze this data, we observed significant increases in proliferation for PTPN22 KO cells using L-TCR at the 2 highest peptide doses while an ANOVA test for the H-TCR titrations failed to identify difference at any peptide dose.

Reviewer #1 (Recommendations for the authors):This reviewer commends the authors on the near Hereculean efforts undertaken to generate the range of genetically modified cord blood T cells. for study.General comments:It might be worth mentioning the specificity of the TCRs in the abstract just to give the story context. It took this reviewer a while to find this info, buried in methods. The Results section reads a bit like a discussion or even review paper, focused on the nuances of gene editing. This reviewer would suggest keeping the narrative of the results focused on describing the experimental findings, and confining technological limitations and challenges to the discussion. This is a somewhat stylistic issue, but would make for a better read.

We agree with the Reviewer’s suggestion. The antigen specificity of the TCRs utilized in our study is now included in the abstract (Lines: 32-33), methods (Lines: 561-564), and results (Lines: 266-267) to provide the requested context.

Also, we have tried to revise the Results section to limit our technical discussion. However, as this remains a highly technical paper, we want to assure that the steps taken are clear to diverse audiences who may find this paper relevant and that hope to design related studies in other cell models.

Reviewer #2 (Recommendations for the authors):Overall, this conclusion is well funded based on the experiments shown, with only a few issues or questions (see below).1) In the figures the different symbols represents different donors, which is clearly marked. However, the data in all experiments are from different experiments, it would be helpful if it could be marked also which symbols belong to which experiment (maybe using different colors) to get a grip on the interexperimental variation ( I understood the values are not normalized?).

The Reviewer is correct that the values are not normalized. Summary flow data is presented as either raw MFI or percent positivity. This approach was used based upon input from coauthors expressing a desire for the data to be presented in as unmanipulated a fashion as possible. If there are key data that the Reviewer feels should be shown in a normalized format, we would be happy to provide that. It should be noted, however, that for all figures if a donor is used, then all gene edited versions of that donor’s cells are also used in the same assay (Example: Figure 2 was generated in 2 experiments, with 2 donors per experiment. For each donor in an experiment, we used all 4 genotypes, PTPN22 wt, non-risk, risk, and KO). This approach was used to control for potential batch effects. The only instance where this is not the case is the cytokine secretion data in the updated figure 5, where we added to previous data with new datapoints and increased genotypes studies to address reviewer requests.

Regarding reformatting figures: As this work was primarily done with cord blood derived samples, which provide variable and limited cell numbers, the use of multiple donors was required. After editing and lentiviral transduction, cells were frozen and banked and some used for multiple experiments, but we had insufficient cell numbers to use the same donors for all experiments. Over the course of the study, we used >10 cord donors. Unfortunately, we do not believe there is a satisfying and clear way to symbolize all different T cell donors simultaneously if that is the desired request. Such labeling would generate a large variety of colors and symbols and complicate what we believe to be an already complex study.

2) The reference list is dominated by reviews. It is important to refer to original publications not only because it is more fair but also to prevent false statements going into circuits. Pls take away most, or preferably all, of these references and replace with original when needed.

We agree with the Reviewer’s comment regarding citing primary data whenever feasible and have revised our study to try to address this request.

In our initial manuscript we referenced 52 publications, 11 of which were review articles, and 2 more were meta-analyses. We elected to retain the following 5 reviews and 2 meta-analyses references:

Refs 1 and 2 were used to introduce and support of our opening statement, “The risk of autoimmunity stems from a complex interplay of genetic and environmental factors.”We retained the meta-analyses to best emphasize broad range of autoimmune diseases associated to PTPN22.The review by Rawlings, Dai, and Buckner was retained as it refers to arguments regarding gaps in understanding of the PTPN22 risk variant, and its relevance to treatment of autoimmunity.Finally, the reviews regarding DNA repair (Scully et al., and Xue et al.) were retained. The cellular responses to DNA breaks is complex; with primary data including hundreds of publications spanning decades, and beyond the scope of this study. These reviews were used to help to provide context on how ssODNs integrate into the DNA of primary cells post Crispr cleavage.

All other reviews have been replaced with references to the primary research.

3) In the discussion regarding different outcomes in mouse versus human it is pointed out that the mouse is in SPF conditions and are genetically more homogenous. This is true but there are likely even more important issues regarding experimental settings. Just to mention taking out T cells from their normal biological context, put them in an incubator with atmospheric oxygen pressure represent a shock and dramatically changes the context. This is of particular importance for PTPs, including PTPN22, which are profoundly redox regulated. There are numerous differences and usually the most important differences are the experimental settings and I think it should be pointed out and discussed in a better way. For future studies, the mouse is absolutely needed, for example, the lymphocyte selection, both T and B, needs to be addressed and I guess experimental mice are critically needed to solve this. Thus, the main conclusion from this paper is that it provides good arguments to improve the animal models to solve the functional genetics of PTPN22 (and most likely other SNPs).

We agree and have emphasized this idea in our discussion (Lines: 444-453) as a critical need for continued use of mouse models in studying this SNP (and pretty much all others).

However, one of the key points we emphasize is that previous in vitro stimulation experiments using T cells from human risk variant carriers vs. R619W KI mice have led to opposing data sets. Yes, the experimental settings for such studies are sub-optimal for PTP function, but such work has none-the-less spurred questioning the propriety of mouse models to study this pathway.

We believe mouse models represent a critical system for studying PTPN22 and show here that in an ex-vivo human model (using genetically homogenous/ environmentally naive primary T cells), a similar phenotype is generated (implying that previous work may be confounded by various factors including environmental, genetic and/or experimental design, that deserve consideration).

We hope our changes have satisfied this point, but if we missed the mark or more elaboration is needed, we are open to revising this further.

Reviewer #3 (Recommendations for the authors):This work employs an experimental system that minimizes human donor heterogeneity and developmental skewing. The authors demonstrate that PTPN22 plays an inhibitory role on TCR regulated effector functions in low, but not high, avidity TCR interactions. The work builds on a report by Salmond et al. (Nature Immunology, 15:875, 2014) using the murine OT-I TCR system that PTPN22 restrains TCR signaling induced by lower affinity agonists (T4 or G4 peptides) than the higher affinity N4 agonist. In the present work, the authors demonstrate further in human T cells that PTPN22 KO or SNP edited cells demonstrate similar enhanced TCR-mediated functions when compared to PTPN22 wildtype T cells. The authors should consider the following:1. As the highest peptide dose was utilized in the signaling studies in Figures 4-6 since this dose produced efficient proliferation, the authors should consider performing a dose-titration with the H-TCR studies to fully support their conclusion that there are no differences in H-TCR signaling in the presence or absence of PTPN22.

We have attempted to address this important comment in our response to major Revision requirements #2 (see above).

2. Page 19, line 439- the level of TCR expression in the transduced cells should be shown.

As stated in the manuscript we utilized an MOI of 5 for gene transfer using TCR expression lentiviral vectors. This approach was used to achieve transduction while limiting the viral copy number per cell allowing for consistent levels of TCR expression. To complete our work, we utilized a series of viral preparations generated within our laboratory and series of human donors. This resulted in a range of transduction rates for both the L-TCR and H-TCR cultures. As an example, the left bar graphs in Author response image 5 show the range of transduction for L-TCR (~5-25%) across 5 human donors based upon expression of the cis-linked GFP marker. Similar results were obtained for the H-TCR. Importantly, lentiviral transduction was performed prior to gene editing; and, as predicted, we observed nearly identical levels of GFP expression and GFP MFI (right bar graphs, Author response image 5) across each editing condition for each donor (each symbol indicates an independent donor). Thus, there is very little variance across different PTPN22 edited cell populations transduced with L-TCR after in vitro expansion and prior to experimental use.

We have revised the description of our LV transduction methods in our methods/results. However, we elected not to add this data (Author response image 5) to our manuscript as we felt that it might be distracting to the information flow. In the results, we updated the text to state: “LV transduction was performed at an MOI of 5 to transduce less than 25% of the culture, with an average of 5 to 15% marking (depending on donor, TCR sequence, and lentiviral lot) designed to ensure low viral copy numbers per transduced cell to achieve similar levels of TCR expression in all transduced cells.” (Lines: 279-283). If Editors or Reviewer prefer, we could add Author response image 5 to our supplemental materials.

**Author response image 5. sa2fig5:** 

If more or different data is requested, we will be happy to provide that to the best of our abilities.

3. The conclusions of the Salmond et al. study mentioned above describing differences in low vs high affinity agonists should be provided greater description as background.

Agreed, we have added more information about that study in the Introduction (Lines: 89-94).

4. The authors should also discuss how their conclusions compare with the recent study of Perry et al. (Journal of Immunology, 207:849, 2021) where using an overexpression system, they conclude that the PTPN22(620W) SNP is hypomorphic in human CD4^+^ T cells rather than a full loss-of-function.

Thank you for bringing this reference to our attention. It is referenced and discussed in revised Discussion section (Lines: 479-493).